# Association between attitudes of stigma toward mental illness and attitudes toward adoption of evidence-based practice within health care providers in Bahrain

**Feras Al Saif**[1]*, **Hussain Al Shakhoori**[1], **Suad Nooh**[1‡], **Haitham Jahrami**[1,2‡]

**1** Psychiatric Hospital, Ministry of Health, Manama, Kingdom of Bahrain, **2** College of Medicine and Medical Sciences, Arabian Gulf University, Manama, Kingdom of Bahrain

☯ These authors contributed equally to this work.
‡ These authors also contributed equally to this work.
* dr.feras.alsaif@gmail.com

**Data Availability Statement:** All relevant data are available from Dryad at DOI:10.5061/dryad.

## Abstract

The health care system is one of the key areas where people with mental illnesses could experience stigma. Clinicians can hold stigma attitudes during their interactions with patients with mental illness. To improve the quality of mental health services and primary care, evidence-based practices should be disseminated and implemented. In this study, we evaluated the attitudes of health care providers in Bahrain toward people with mental illness and adoption of evidence-based practice using the Opening Minds Stigma Scale for Healthcare Providers (OMS-HC) and Evidence-Based Practice Attitude Scale (EBPAS). We conducted a cross-sectional study across 12 primary health care centers and a psychiatric hospital (the country's main mental health care facility). A self-report questionnaire was distributed among all health care providers. A total of 547 health care providers participated, with 274 from mental health services and 273 from primary care services. Results of the OMS-HC indicated differences between both main groups and subgroups. Regression model analysis reported significant outcomes. There was no statistical difference found between both groups in EBPAS scores. A weak but statistically significant negative association was reported between both scales. Participants showed varying stigma attitudes across different working environments, with less stigma shown in mental health services than in primary care services. Providers who were more open to adopting evidence-based practices showed less stigma toward people with mental illness. Comparing our findings with previous research showed that health care providers in Bahrain hold more stigma attitudes than other groups studied. We hope that this study serves as an initial step toward future campaigns against the stigma of mental illness in Bahrain and across the region.

4b6v2f0. URL: https://datadryad.org/stash/dataset/
doi:10.5061/dryad.4b6v2f0.

**Funding:** The authors received no specific funding
for this work.

**Competing interests:** The authors have declared
that no competing interests exist.

# Introduction

Stigma has been identified as a mark of shame, disgrace, or disapproval which results in an individual being rejected, discriminated against, or excluded from participating in a range of areas within society [1]. People with mental illness are often stigmatized in a way that could be considered a "secondary illness," after being labeled with a mental health disorder [2]. Rush et al. argued that people with mental illness often first experience the problem of coping with their mental illness, then a lack of understanding from society regarding mental health disorders, which often results in stigma [3].

The stigma of mental illness is a complex concept [4, 5], with multiple aspects and subcomponents that have been identified in previous studies. These domains include "perceived stigma" [6–9], "public stigma" [9], "self-stigma" [10], "structured stigma" [11–13], "social distance" [7, 14], "dangerousness" [6], "recovery" [15], "emotional reactions" [16], and "social responsibility and compassion" [17]. Another type of stigma often associated with mental illness is "double stigma," which is stigma compounded by membership in more than one stigmatized group, such as due to one's HIV/AIDS status, identifying as LGBT, ethnicity, religion, etc. [13]. These domains are important in assessing mental health care services [9, 18]. A comprehensive assessment should include three components of stigma formation according to the social-cognitive model; Stereotype (cognitive), Prejudice (affective), and Discrimination (behavior) [13].

Stigma of people with mental illness can be a major barrier to help-seeking behavior, access to care resources, or life opportunities (e.g., not receiving work or housing, being underserved, or being socially marginalized) [2, 9, 19–32]. Stigma can also damage campaigns that advocate for mental health treatment and, as a consequence, fewer people with mental illness are diagnosed and treated [1]. These effects greatly impact the quality of life for people with mental illness, as well as can cause risks to and burdens on the public [33, 34]. People with mental illness have been shown to have poor health outcomes and even premature death [35]. The reasons for such complications vary; however, research has shown that medical professionals can misattribute physical symptoms in people with mental illness as being part of their mental health condition, a phenomenon known as "diagnostic overshadowing" [13, 36–38]. Thus, medical conditions could be overlooked, leading to a lack of proper treatment, possibly due to a health care provider's discriminatory attitude [38–41]. Another contributing factor could be that some clinicians are not well trained regarding psychiatric issues [38, 42].

Health care providers are frequently seen in the public eye and portrayed in the media as having positive attitudes, being empathic toward people with mental illness, and offering compassionate care; however, this is often not the case [43, 44]. The health care system is one of the key areas where people with mental illness could experience stigma [2, 13, 25]. Numerous studies on this topic suggested that health care professionals hold some form of stigma against people with mental illness, often influenced by the media [2, 25]. Psychiatrists also often stigmatize their own patients with mental illness during their daily interactions or when reviewing their symptoms and side effects of medications, such as drooling, obesity, and tardive dyskinesia, and these patients are often regarded as undesirable or dangerous [2, 25, 45]. Stigma does vary depending on differences in mental health disorders and health care professionals. Mental health professionals can be subjected to stigma as well, due to their close contact with stigmatized groups, also known as "stigma by courtesy" [13, 25, 46].

Education alone was found not to reduce the stigma of mental illness [47]. Numerous strategies have been proposed to challenge the stigma of mental illness within the health care system, using models of stigma to reduce misinformation, prejudice, and discriminatory behaviors [47–51]. An interesting approach emerged emphasizing the role of a contact-based

educational program in reducing stigma [47, 50, 52–55]. This strategy provides a program within a mental health curriculum that allows mentally ill patients to share their own experiences with their mental illness with trainees and health care providers who may hold stigmatizing views [55, 56].

Better mental health care should be supported by ethics, evidence, and experience [57]. Unfortunately, in clinical practice, mental health care is often not based on evidence of efficacy or effectiveness [58, 59]. To improve the quality of mental health services and care in real-world settings, evidence-based practices (EBPs) should be disseminated and implemented [60, 61]. A recommended approach would be through realignment of the evidence base with clinical practice and healthcare service[58]. Studies reported that the use of EBPs improve mental health care and help reduce stigma held by mental health professionals [57, 62]. A Japanese cross-sectional study conducted among rehabilitation psychiatric staff reported that EBP experience is associated with low individual levels of stigmatization [62].

To our knowledge, there is only one study in Bahrain that assessed the attitude of primary health care physicians toward mental illness, based on a self-report questionnaire [63]. The researchers utilized 25 statements to assess this attitude, and found that the majority of primary health care physicians had a favorable attitude toward people with mental illness [63]. Interestingly, only those with psychiatry qualifications or certificates reported the most favorable attitudes [63]. There were several limitations of this study, however, as it lacked a validated tool to measure stigma attitudes and the scope of the population studied was limited [63]. Thus, little is known about the stigma attitudes of health care providers toward people with mental illness in Bahrain, or the relationship between stigma and health care providers' attitudes toward adopting EBPs.

Therefore, we conducted this study with two goals in mind. First, we aimed to perform an exploratory investigation utilizing a validated tool to assess the level of stigma of mental illness among health care professionals working in mental health and primary care services in the Kingdom of Bahrain. Second, we examined whether there is an association between attitudes toward adopting EBPs and attitudes of stigma toward mental illness. We hypothesized that health care professionals who work in close contact with patients with mental illness would show lower scores, and those with more positive attitudes toward adopting EBPs would display less stigma.

## Materials and methods

### Study design, participants, and data collection

This study had a cross-sectional design. The study participants were health care professionals licensed to practice in the Kingdom of Bahrain, in accordance with the National Health Regulatory Authority (NHRA), who provided written consent to participate in the study [64]. The studied sample was grouped into two major categories. The first included participants working in a psychiatric hospital (PSYCH), which is the only government-sponsored hospital specializing in mental health care that provides secondary and tertiary mental health services to residents of the Kingdom. The second included participants working in Primary Healthcare Centers (PHCs), which provide preventive or curative health care services to families and communities and are spread over the country's four governorates [65].

### Sample size estimation and sampling methods

An estimated sample size of 385 was calculated using the epidemiology equation with a confidence level of 95%, a prevalence rate of 50%, a precision level of 5%, and an infinite assumption population. We sampled 12 out of 24 registered active centers in Bahrain, and the

selection process was organized to sample centers from every governorate of the country, proportional to the catchment area of each regional density population [65]. Convenience sampling of the study's population was performed through two stages. The first stage was directed toward all PSYCH staff from almost every department (mental health care physicians, nurses, psychologists, physiotherapists, and social workers), excluding only the administration and pharmacy departments. The second stage included the selection of PHC staff (primary health care physicians, nurses and social workers). The total duration of the recruitment period for the whole sample was seven months, starting in August, 2017 and ending in February, 2018.

### Pilot study

A pilot study was performed prior to sampling to test feasibility. A total of 10 randomly selected participants were collected from the accident and emergency department of Salmanyia Medical Complex, the main secondary and tertiary government-sponsored hospital in the country. There were no issues reported in comprehension or answering the suggested questionnaire [65].

### Measures

The instrument used in this study was a self-report questionnaire with three sections. The language used was English, the country's second most widely used language after Arabic, and the official language used in medical universities, clinical training programs, and the health care profession [66]. The first section collected sociodemographic data such as age, gender, nationality, professional ranking, and years of experience.

The second section was adapted from the 15-item Open Mind Stigma Scale for Health Care Providers (OMS-HC) that measures health care providers attitudes toward people with mental illness [4]. The original version of the OMS-HC contains 20-items and was developed by Kassam et al. [43]. The corresponding author of the OMS-HC was contacted and permission to use this scale in our study was obtained. We chose to use the standardized 15-item version, as recommended in a study which examined the scale's properties [4]. This scale was reduced to 15-items because of weak item-total correlation (below 0.20) found in four items and additional item was dropped after cross-loaded across all three factors equally; ending with the 15-item version[4]. It comprises three subscales: the first subscale is regarding the attitude of health care providers toward people with mental illness, the second is regarding disclosure/help-seeking, and the third is regarding social distance [4]. The internal consistency of the 15-item OMS-HC ($\alpha$ = 0.79) and its three subscales ($\alpha$ = 0.67–0.68) were reported to be acceptable [4]. Each item of this scale has scores ranging from 1 to 5, and the full 15-item OMS-HC score can range from 15 (least stigmatizing) to 75 (most stigmatizing) [4]. A recent systematic review of all available instruments that assess mental health related stigma among health care professionals concluded there is no "gold standard" tool; however, the 15-item OMS-HC was identified as the strongest available tool supported by evidence [67]. However, another recent multi-center study published online which investigated the validity of this scale, utilizing Rasch modeling, with undergraduate nursing students did not support its use as a global estimate of stigma [68]. Conclusions made from this study would have been more credible has it taken into account the views of healthcare professionals in addition to undergraduate students.

The third section was composed of the 15-item Evidence-Based Practice Attitude Scale (EBPAS) [59]. The EBPAS was developed to assess mental health providers' attitudes toward the use of innovation and EBPs in mental health settings [69, 70]. The corresponding author for the EBPAS was contacted and provided permission to use this scale in our study. The

EBPAS has shown good reliability and validity, and been used widely in the United States [71–73]. It comprises four subscales assessing appeal, requirements, openness, and divergence. For each subscale, a mean is calculated, with a total score composed of the mean of all items calculated [59]. The EBPAS is scored using a 5-point Likert scale [59]. Higher scores indicate a more positive attitude toward the adoption of EBPs [59]. The EBPAS was selected for its ability to assess a wide range of domains that influence health care professionals when applying the best available EBPs [59, 70].

### Data collection, procedure, and ethical approval

All PSYCH and PHC staff were invited to participate in this study. The participants were voluntarily involved and informed about protection of confidentiality, and did not receive any commercial benefits from participation. They were provided with information regarding the main aim of the study and had the right to withdraw at any time. Written consent was provided before answering the questionnaire. The estimated duration of time to complete the questionnaire was no more than 10 minutes. Only those who voluntarily consented to participate in the study and completed the questionnaire were included (response rate: 80.1%). We excluded 26 questionnaires with significant amounts of missing data. This study was approved by the Secondary Health Care Research Sub-committee of the Ministry of Health in the Kingdom of Bahrain, in meeting No. 08/17, held on April 7, 2017.

### Statistical analyses

We used SPSS version 25.0 to analyze our data. Missing values analysis (MVA) was performed prior to any testing, and the MVA results revealed that missing values were not a significant issue, as they occurred completely at random. Descriptive statistics, including means and standard deviations (SD), were calculated for continuous variables, and frequencies and counts were calculated as point estimates for the sample. In addition, 95% confidence intervals (CI) were calculated to present an estimate of the variation around the point estimates. Cronbach's alpha was used to estimate the reliability coefficient properties of the scales used in this study. A factor analysis technique was used to examine the validity of the scales, before performing factor analysis using the Kaiser-Meyer-Olkin (KMO) measure of sampling adequacy (MSA), and Bartlett's test of sphericity was used to mathematically establish suitability for conducting a factor analysis. The MSA index in our study was 0.9 (reference range 0 to 1.0). Additionally, Bartlett's test of sphericity revealed results of $p = 0.001$, indicating there was a high probability of significant relationships between the variables.

Exploratory factor analysis was carried out using the maximum likelihood extraction (MLE) technique. The MLE method was used because of its known sensitivity in factor extraction, as it produces parameter estimates that are the most likely to have produced the observed correlation matrix. Promax rotation was used to ease the interpretation of the solution, and the oblique type rotation was used to allow the factors to be correlated with each other. To examine differences between groups, independent sample $t$-tests (for continuous data) or Chi-square tests (for categorical data) were utilized to compare subgroups. A p-value $< 0.05$ was considered to be statistically significant. A linear regression model was used to investigate the association between the OMS-HC and predictive/independent variables group, age, gender, nationality, professional background, and experience.

# Results

## Participant characteristics

The sample included 547 health care professionals who consented to participate in the study; 274 were from PSYCH and 273 were from PHCs. There was variation in the response rates between both groups, as 317 questionnaires were sent to PSYCH (response rate 86.4%), while 361 questionnaires were sent to PHCs (response rate 75.6%). The total sample response rate was 80.1%. Only data from complete questionnaires were included in the sample. Demographic characteristics of the sample are summarized in Table 1.

The majority of participants were younger than 44 years old (82%), and the age range was statistically nonsignificant across both groups ($\chi^2$ = 4.745, $df$ = 3, $p$ = 0.191). The number of females was significantly higher than males ($\chi^2$ = 68.948, $df$ = 1, $p$ < 0.001), with 77% of the total sample. This gender difference was noted across both groups, and 91.9% of the participants working in PHCs were female, compared to only 62% working in PSYCH. The total percentage of Bahraini participants was reported to be 62.3%. There were significant differences between professional groups and rankings ($\chi^2$ = 30.956, $df$ = 9, $p$ < 0.001). The majority of the study's participants were nurses (79.2%), while physicians across both specialties (mental health or primary care) totaled 16.4%. The remaining Allied Health Professionals, which constitutes psychologists, social workers, and occupational therapists, were 0.04% of the total sample. It should be noted that professional ranking was based on NHRA regulations for health care professionals [64]. Those who obtained board-certified specialty qualifications were reported as specialist physicians, while those who did not, but were still working, were listed as service physicians. It is worth mentioning that 74.2% of our study's participants had been working within the health care system for more than five years. There was a significant difference between years of experience in the two groups ($\chi^2$ = 13.835, $df$ = 2, $p$ = 0.001).

## Opening Minds Scale for Health Care Providers

Sum scores and SDs of the 15-item OMS-HC for each variable are shown in Table 1. There were 55 participants who did not complete all 15-items, and were thus excluded from analysis. The sum score of participants in PSYCH was 39.3, which was statistically significant compared to those in PHCs, who had a sum score of 42.5 ($p$ < 0.001). Scores from PSYCH were reported to be lower than from PHCs for almost all variables. For participants under 44 years old, there were statistically significant differences ($p$ < 0.001) between both groups. Statistically significant differences were shown among female participants ($p$ < 0.001), but not among males ($p$ < 0.635). The nationality of participants was a significant factor that had an impact on sum scores. Bahrainis at PSYCH were the subgroup least stigmatized by others, with a sum score of (38.4), compared to their colleagues in PHCs ($p$ < 0.001). Comparing the sum scores within each group between Bahrainis and non-Bahrainis in PSYCH or PHCs showed statistically significant differences ($p$ = 0.02, $p$ = 0.003, respectively; not shown in Table 1).

The total physicians' scores showed no significant differences across both groups ($p$ = 0.146), while only the subgroup of the service physicians showed a statistically significant difference ($p$ = 0.040). Nurses in PSYCH, compared to PHCs, showed statistically significant differences ($p$ < 0.001), with a sum score difference of -3.6. Years of experience had a remarkable impact on the total sum score of participants, showing statistically significant differences for those who worked more than five years ($p$ < 0.001); however, it was not statistically significant for those with less than one year of experience or between one and five years ($p$ = 0.556, $p$ = 0.051, respectively). The sum scores for the subscales of attitudes of health care providers toward people with mental illness and disclosure/help-seeking were statistically significant

**Table 1. Demographic characteristics and 15-item OMS-HC.**

| Demographic characteristics | PSYCH | 15-item OMS-HC | | PHCs | 15-item OMS-HC | | All | 15-item OMS-HC | | Independent t-test and *p* value (bold means statistically significant) |
|---|---|---|---|---|---|---|---|---|---|---|
| | n(%) | Sum | SD | n(%) | Sum | SD | n(%) | Sum | SD | |
| **Total** | 274 (100%) | 39.3 | 6.6 | 273 (100%) | 42.5 | 6.2 | 547 (100%) | 40.9 | 6.6 | **< 0.001** |
| **AGE** | | | | | | | | | | |
| < 34 | 103(37.7) | 38.1 | 6.9 | 126(46.1) | 41.8 | 6 | 229(41.9) | 40.1 | 6.6 | **< 0.001** |
| 35–44 | 114(41.8) | 40 | 6.3 | 105(38.5) | 43.7 | 6.3 | 219(40.1) | 41.8 | 6.6 | **< 0.001** |
| 45–54 | 49(17.9) | 39.8 | 6.5 | 36(13.2) | 41.7 | 6.3 | 85(15.6) | 40.6 | 6.4 | 0.214 |
| > 55 | 7(2.6) | 40.2 | 8.6 | 6(2.2) | 41.8 | 9.9 | 13(2.4) | 40.9 | 8.6 | 0.813 |
| **Gender** | | | | | | | | | | |
| Female | 170(62) | 38.8 | 6.6 | 251(91.9) | 42.8 | 6 | 421(77) | 41.2 | 6.5 | **< 0.001** |
| Male | 104(38) | 40 | 6.7 | 22(8.1) | 39.2 | 7.9 | 126(23) | 39.9 | 6.9 | 0.635 |
| **Nationality** | | | | | | | | | | |
| Bahraini | 150(54.9) | 38.4 | 6.9 | 190(69.6) | 41.8 | 6.6 | 340(62.3) | 40.3 | 6.9 | **< 0.001** |
| Non-Bahraini | 123(45.1) | 40.4 | 6.1 | 81(29.7) | 44.2 | 4.9 | 204(37.4) | 41.8 | 5.9 | **< 0.001** |
| **Professional & Ranking** | | | | | | | | | | |
| *Physician*: | 39(14.2) | 36.8 | 7.3 | 51(18.6) | 39.2 | 7.5 | 90(16.4) | 38.1 | 7.4 | 0.146 |
| -Consultant | 9(3.3) | 36.4 | 6.8 | 13(4.8) | 38 | 8.8 | 22(4) | 37.3 | 7.8 | 0.671 |
| -Specialist | 10(3.6) | 35.7 | 7.5 | 7(2.6) | 40.8 | 6.6 | 17(3.1) | 37.7 | 7.4 | 0.197 |
| -Service | 7(2.6) | 37.3 | 2.9 | 2(0.7) | 44.5 | 4.9 | 9(1.6) | 39.1 | 4.5 | **0.040** |
| -Trainee | 13(4.7) | 37.6 | 9.3 | 29(10.6) | 38.9 | 7.3 | 42(7.7) | 38.5 | 7.9 | 0.669 |
| *Nurse*: | 217(79.2) | 39.9 | 6.4 | 216(79.1) | 43.5 | 5.5 | 433(79.2) | 41.7 | 6.3 | **< 0.001** |
| -Head | 7(2.6) | 38 | 11 | n/a | | | 7(1.3) | 38 | 11 | n/a |
| -Supervisor | 14(5.1) | 38.5 | 6.6 | 7(2.6) | 38.3 | 3.3 | 21(3.8) | 38.4 | 5.8 | 0.955 |
| -Staff | 196(71.5) | 40 | 6.2 | 209(76.6) | 43.7 | 5.5 | 405(74) | 41.9 | 6.1 | **< 0.001** |
| *Allied Health Professionals* | 18(0.7) | 38.3 | 6.9 | 6(0.2) | 36.3 | 3.7 | 24(0.4) | 37.8 | 6.2 | 0.516 |
| -Psychologist | 4(1.5) | 33.2 | 7.7 | n/a | | | 4(0.7) | 33.2 | 7.7 | n/a |
| -Social Worker | 7(2.6) | 39.7 | 0.8 | 6(2.2) | 36.3 | 3.7 | 13(2.4) | 38 | 3.1 | 0.058 |
| -Occupational therapist | 7(2.6) | 40 | 8.5 | n/a | | | 7(1.3) | 40 | 8.5 | n/a |
| **Years of experience** | | | | | | | | | | |
| < 1year | 10(3.7) | 40.6 | 4.1 | 23(8.4) | 42 | 6.7 | 33(6) | 41.6 | 6.1 | 0.556 |
| 1–5 years | 42(15.4) | 38.8 | 7 | 66(24.2) | 41.5 | 6.3 | 108(19.8) | 40.4 | 6.7 | 0.051 |
| > 5years | 221(81) | 39.3 | 6.7 | 184(67.4) | 43 | 6.1 | 405(74.2) | 40.9 | 6.7 | **< 0.001** |

Variables of age, gender, nationality, professional group, ranking, and years of experience are listed comparing both groups sum scores of the 15-item OMS-HC.

(*p* < 0.001), but the subscale for social distance showed no statistically significant difference (*p* = 0.402; not shown in Table 1). The internal consistency of the total 15-item OMS-HC was analyzed, and Cronbach's was alpha reported to be 0.6, which was interpreted as questionable [74, 75].

## Stigma regression model

A linear regression model was used with OMS-HC scores as the dependent variable, and group, age, gender, nationality, professional background, and experience as independent variables. Results indicated that group ($\beta$ = 0.283, *p* = 0.001), age ($\beta$ = 0.109, *p* = 0.035), nationality ($\beta$ = 0.108, *p* = 0.023), and professional background ($\beta$ = 0.181, *p* = 0.001) were statistically

**Table 2. Stigma regression model results.**

| Model Summary | | | | | |
|---|---|---|---|---|---|
| **Model** | **R** | **R Square** | **Adjusted R Square** | **Std. Error of tde Estimate** | |
| | 0.338[a] | 0.114 | 0.107 | 6.26839 | |
| **ANOVA** [b] | | | | | |
| **Model** | **Sum of Squares** | **df** | **Mean Square** | **F** | **Significant(bold means statistically significant)** |
| **Regression** | 2449.914 | 4 | 612.479 | 15.588 | **0.001**a |
| **Residual** | 19056.986 | 485 | 39.293 | | |
| **Total** | 21506.9 | 489 | | | |
| **Coefficients**[b] | | | | | |
| **Model** | **Unstandardized Coefficients** | | **Standardized Coefficients** | **t** | **Significant(bold means statistically significant)** | **95.5% Confidence Interval for B** | |

| **Model** | **B** | **Std. Error** | **Beta** | **t** | **Significant(bold means statistically significant)** | **Lower Bound** | **Upper Bound** |
|---|---|---|---|---|---|---|---|
| (Constant) | 28.05 | 1.732 | | 16.192 | **0.001** | 24.647 | 31.454 |
| group | 3.654 | 0.575 | 0.276 | 6.358 | **0.001** | 2.525 | 4.783 |
| age | 0.772 | 0.383 | 0.09 | 2.014 | **0.045** | 0.019 | 1.525 |
| nationality | 1.333 | 0.624 | 0.098 | 2.136 | **0.033** | 0.107 | 2.558 |
| Professional background | 0.665 | 0.173 | 0.175 | 3.851 | **0.001** | 0.326 | 1.004 |

A linear regression model was used to investigate the association between OMS-HC scores and the predictive/independent variables of group, age, gender, nationality, professional background, and experience. A refined regression model was reported excluding the non-significant variables of gender and experience.

[a]Predictors: (Constant), group, age, nationality, professional background.

[b]Dependent Variable: OMS-HC

significant predictors, controlling for other variables while gender (β = -0.031, p = 0.509) and experience (β = -0.046, p = 0.385) were not statistically significant. A refined model was analyzed limited toward these significant predictors group, age, nationality, and professional background and excluded non-significant predictors gender and experience. The place of work showed the highest effect on stigmatization compared to other factors (β = 0.276, p = 0.001). Its effect is three times higher than the effect of age (β = 0.09, p = 0.045) or nationality (β = 0.098, p = 0.033) and almost twice the effect of professions (β = 0.175, p = 0.001). The collective effect of these variables was moderate as Adjusted R Square reported to be 0.107, Std. Error of the Estimate was 6.26. Details of the results are presented in Table 2.

## Evidence-Based Practice Attitude Scale

The means and SD of the 15-item EBPAS scores from every group and subgroup are shown in Table 3.

Mean EBPAS scores for PSYCH were reported to be 2.39, which was not statistically significant, compared to the mean score for PHCs of 2.41 (p = 0.201). There were no significant differences in means across all other variables. The internal consistency of the total 15-item EBPAS was analyzed, and Cronbach's alpha was reported to be 0.703, which was interpreted as acceptable [74, 75].

## Association

A scatter plot created to explore an association between full scores for both scales. We used Pearson's product correlation coefficient (-0.256, p < 0.001), as displayed in Fig 1.

## Discussion

To our knowledge, this is the first study in Bahrain that compares attitudes of stigma toward mental illness among health care providers from different health care settings. In this study, we reported a statistically significant negative association does exist between both attitude scales of stigma toward mental illness and the adoption of EBPs. Our study findings are in line with most previous studies that reported health care providers do express attitudes of stigma toward people with mental illnesses [45, 76–79]. Psychiatrists have been reported to stigmatize their patients [45, 77, 80, 81], as have general practitioners, which can have an impact on the treatment received by patients with chronic mental illnesses [81, 82].

In our study, we found that those who were in contact more frequently with patients with mental illness, as seen in the PSYCH group, expressed fewer attitudes of stigma than those in the PHC group. Several previous studies have highlighted the importance of a contact-based approach to reducing stigma [51, 52, 81, 83–90]. This approach could be explained through the contact hypothesis [91]. This means that by increasing the personal and professional interactions with mentally ill people more positive attitudes and less stigma develop [83, 84, 89].

Further, our study found that health care professionals younger than 44 years old showed fewer attitudes of stigma than older professionals. These findings contradict the findings of some previous studies which indicated that younger individuals and students exhibited more negative attitudes [44, 79, 92]. However, in a transcultural study comparing stigma attitudes in Switzerland and Brazil, it was reported that younger individuals showed less social distance toward people with mental illness [91, 93, 94]. Previous studies have also indicated gender variations, as females were reported to show fewer stigma attitudes than males [44, 79, 80, 90, 95, 96]. In our study, it was found that females in PSYCH were the subgroup with the fewest stigma attitudes. This could be explained by the tendency of females to be more empathetic toward people with mental illness [44, 97–99].

We also found that Bahrainis working in PSYCH or PHCs showed fewer stigma attitudes. We believe there could a communication barrier leading health care professionals to misunderstand patients' suffering and prevent an effective physician-patient relationship. This relationship should include an understanding of patients' problems, concerns, and expectations, and physicians being more engaged, empathetic, and supportive could help patients express themselves more fully and perceive discussions with health care providers in a more meaningful way [100]. It was reported that a lack of attention to language and culture for users of mental health services can prevent the establishment of adequate communication and trust, and act as a barrier between clients and clinicians [101, 102]. These barriers could explain the differences in stigma scores among individuals from different cultural backgrounds working within the same settings as other health care professionals, who do not speak a patient's native language, which in our case was Arabic [66].

Our data did reveal differences in stigma scores among professionals from different settings. The professional groups in PSYCH which showed the most stigma attitudes were the staff nurses and occupational therapists, while from PHCs, it was the service physicians and staff nurses. Psychologists showed the least amount of stigma attitudes, as was also reported in a previous Swiss study [80]. We believe that awareness about such stigma attitudes among professionals would help rather than harm, and an open discussion would help resolve a problem that is often ignored [80].

Our study did not include a sample from the general public; however, it been reported that health care providers showed less stigma toward people with mental illness, which could be attributed to their professional experience and knowledge of mental health care [79, 91, 103]. However, other studies reported contradictory findings that health care providers showed

**Table 3. Study sample demographic characteristics and 15-item EBPAS scores.**

| Demographic characteristics | PSYCH | | | PHCs | | | All | | | |
| | 15-item EBPAS | | | 15-item EBPAS | | | 15-item EBPAS | | | Independent *t*-test and *p* value |
| | n(%) | Mean | SD | n(%) | Mean | SD | n(%) | Mean | SD | |
|---|---|---|---|---|---|---|---|---|---|---|
| **Total** | 274(100%) | 2.39 | 0.48 | 273(100%) | 2.44 | 0.5 | 547(100%) | 2.41 | 0.49 | 0.201 |
| **AGE** | | | | | | | | | | |
| < 34 | 103(37.7) | 2.43 | 0.5 | 126(46.1) | 2.54 | 0.49 | 229(41.9) | 2.49 | 0.5 | 0.107 |
| 35–44 | 114(41.8) | 2.32 | 0.48 | 105(38.5) | 2.37 | 0.43 | 219(40.1) | 2.35 | 0.46 | 0.535 |
| 45–54 | 49(17.9) | 2.47 | 0.47 | 36(13.2) | 2.36 | 0.6 | 85(15.6) | 2.42 | 0.53 | 0.336 |
| > 55 | 7(2.6) | 2.22 | 0.44 | 6(2.2) | 2.29 | 0.66 | 13(2.4) | 2.25 | 0.52 | 0.843 |
| **Gender** | | | | | | | | | | |
| Female | 170(62) | 2.39 | 0.47 | 251(91.9) | 2.44 | 0.48 | 421(77) | 2.42 | 0.48 | 0.271 |
| Male | 104(38) | 2.4 | 0.51 | 22(8.1) | 2.5 | 0.64 | 126(23) | 2.42 | 0.54 | 0.38 |
| **Nationality** | | | | | | | | | | |
| Bahraini | 150(54.9) | 2.49 | 0.5 | 190(69.6) | 2.5 | 0.53 | 340(62.3) | 2.49 | 0.52 | 0.842 |
| Non-Bahraini | 123(45.1) | 2.27 | 0.43 | 81(29.7) | 2.33 | 0.38 | 204(37.4) | 2.29 | 0.41 | 0.359 |
| **Professional & Ranking** | | | | | | | | | | |
| *Physician*: | 39(14.2) | 2.49 | 0.53 | 51(18.6) | 2.64 | 0.57 | 90(16.4) | 2.57 | 0.55 | 0.199 |
| -Consultant | 9(3.3) | 2.43 | 0.41 | 13(4.8) | 2.7 | 0.64 | 22(4) | 2.59 | 0.56 | 0.29 |
| -Specialist | 10(3.6) | 2.72 | 0.58 | 7(2.6) | 2.67 | 0.62 | 17(3.1) | 2.7 | 0.57 | 0.846 |
| -Service | 7(2.6) | 2.47 | 0.46 | 2(0.7) | 1.94 | 0.27 | 9(1.6) | 2.35 | 0.47 | 0.173 |
| -Trainee | 13(4.7) | 2.36 | 0.59 | 29(10.6) | 2.65 | 0.53 | 42(7.7) | 2.56 | 0.56 | 0.114 |
| *Nurse*: | 217(79.2) | 2.37 | 0.48 | 216(79.1) | 2.39 | 0.46 | 433(79.2) | 2.38 | 0.47 | 0.562 |
| -Head | 7(2.6) | 2.59 | 0.41 | N/A | | | 7(1.3) | 2.59 | 0.41 | N/A |
| -Supervisor | 14(5.1) | 2.74 | 0.65 | 7(2.6) | 2.49 | 0.63 | 21(3.8) | 2.66 | 0.64 | 0.418 |
| -Staff | 196(71.5) | 2.33 | 0.46 | 209(76.6) | 2.39 | 0.46 | 405(74) | 2.36 | 0.46 | 0.206 |
| *Allied Health Professionals* | 18(0.7) | 2.45 | 0.4 | 6(0.2) | 2.58 | 0.67 | 24(0.4) | 2.49 | 0.47 | 0.571 |
| -Psychologist | 4(1.5) | 2.32 | 0.32 | N/A | | | 4(0.7) | 2.32 | 0.32 | N/A |
| -Social Worker | 7(2.6) | 2.45 | 0.41 | 6(2.2) | 2.58 | 0.67 | 13(2.4) | 2.51 | 0.52 | 0.667 |
| -Occupational therapist | 7(2.6) | 2.53 | 0.48 | N/A | | | 7(1.3) | 2.53 | 0.48 | N/A |
| **Years of experience** | | | | | | | | | | |
| < 1year | 10(3.7) | 2.32 | 0.42 | 23(8.4) | 2.51 | 0.52 | 33(6) | 2.45 | 0.5 | 0.339 |
| 1–5 years | 42(15.4) | 2.38 | 0.49 | 66(24.2) | 2.57 | 0.54 | 108(19.8) | 2.49 | 5.3 | 0.083 |
| > 5years | 221(81) | 2.4 | 0.49 | 184(67.4) | 2.39 | 0.47 | 405(74.2) | 2.4 | 0.48 | 0.964 |

Study sample's variables of age, gender, nationality, professional and ranking, and years of experiences are listed comparing both groups scores of 15-item EBPAS mean and SD.

more stigma attitudes than the general population [78, 80, 104]. These findings were not generalized, but were interpreted in accordance with sample characteristics, cultural background of participants, burnout tendencies of practitioners, and the type of mental health disorders being investigated. However, it can be agreed that these findings do strengthen the view that both health care providers and the general public share a common attitude of stigmatization toward people with mental illness [78–80, 103, 104].

The impact of work experiences on stigma has been previously reported.[81, 86, 105, 106] In our study, we found that the duration of work experience (more than five years) significantly affected scores related to stigma. These findings could also highlight the importance of work experience on stigma attitudes toward mental illness, which was reported to show a greater effect than education in itself [90, 107]. Some would assume a paradox, since some

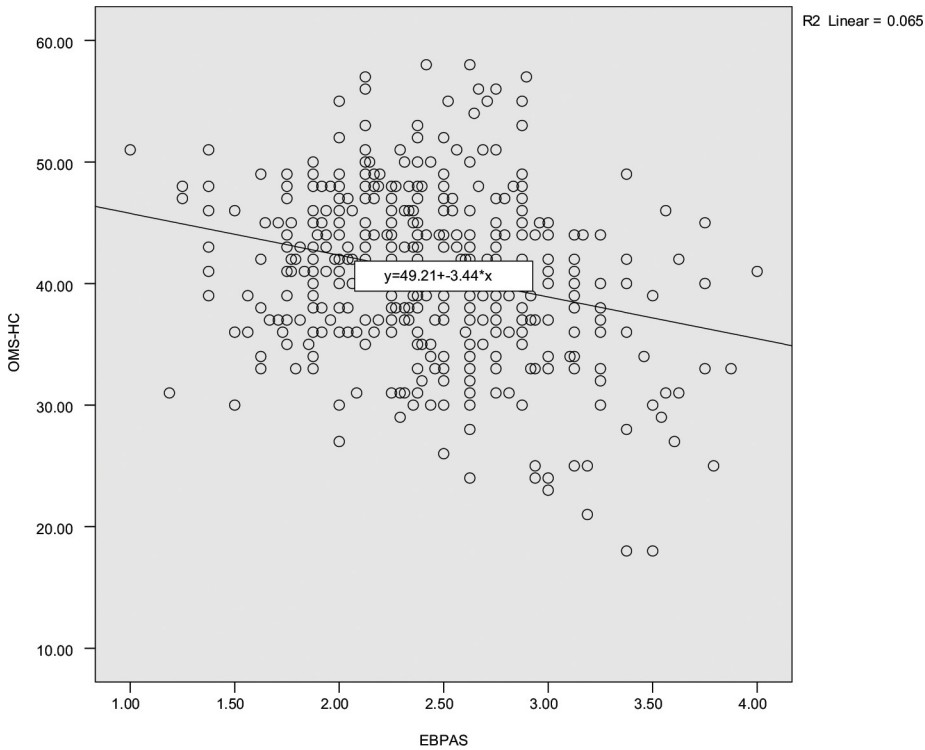

**Fig 1. Scatter plot analyzing total scores of OMS-HC and EBPAS of the study sample.** SPSS analysis of bivariate correlations between EBPAS (shown in X axis) and OMS-HC (shown in Y axis) scores revealed a regression equation and a weak negative association. SPSS revealed the regression equation of analysis (y = 49.21+-3.44*x) and reported that $R^2$ Linear = 0.065. The association was weak in strength and negatively related.

studies further indicated that when health care providers had a personal experience with a mental health disorder, they tended to be react more negatively toward psychiatric patients, which could be understood as the presence of self-stigma [10, 44]. This internal self-concept develops from the recognition of public stereotypes [108]. It involves three steps, the first is the awareness of the public stigma "People with depression are lazy", the second is to agree with such statements "Yes, that's true—depressed people are lazy", and the final step through apply-ing them into their self "I have depression, so I'm lazy" [13, 109–112]. This form of stigma was reported to lower self-esteem and impair an optimistic view on the prognosis of mental illness among family physicians as been noticed in an early study in Bahrain [63, 113, 114].

Based on our results, we believe that those who plan to study/reduce the stigma of mental illness held by health care providers should emphasize the nature of contact with people with mental illness, engage more with all age groups with different professional backgrounds and experience levels, and improve communication to help professionals in understanding patients' suffering, thus facilitating empathy and significantly reducing stigma.

There were only 13 studies reported in the literature that used the 15-item OMS-HC, with different populations and for various aims. A brief literature review of these studies is provided in S1 Table [4, 5, 44, 68, 90, 91, 115–121]. Comparing the OMS-HC scores in our study with similar studies indicated health care providers in Bahrain had more attitudes of stigma than many other groups. The 15-item OMS-HC helped us set a baseline score of stigma attitudes of health care providers in our population and paved the way for future important interventions and follow-up studies [4, 43].

There have been different campaigns against stigma all over the world [13]. Opening Minds, established by the mental health commission of Canada in 2009, is the largest systematic anti-stigma campaign in Canada [122]. It mainly targets youth, health care providers, media, and workplaces through the use of a contact-based education program [13, 123]. There have also been several interventions studied that were developed to fight stigma among health care providers; however, three types are recommended. First is "intensive social contact," which involves encouraging participants to engage with patients with mental illness to break the barriers between "us and them" [124]. Second, the "anti-stigma workshop" provides educational lectures to selected audiences [125]. Third is the "skills-based approach," through which communications skills are improved among health care providers by teaching "what to say" and "what to do" to help patients with mental illness [13]. Furthermore, the help of the media in fighting stigma would certainly be a useful tool, as it is one of the primary sources of the public's awareness of mental illness [126, 127].

## Conclusions

### Final remarks and recommendations

Our study did show that health care providers working in Bahrain expressed significant attitudes of stigma toward people with mental illnesses. Those in frequent contact with people with mental illness showed fewer stigma attitudes, and a weak negative correlation did exist between both attitudes of stigma toward mental illness and adoption of EBPs. We hope our findings serve to help in the fight to further reduce stigma attitudes toward people with mental illness. More research is recommended to investigate views regarding stigma from the general public, other health care providers from different specialties, and the patients themselves.

We recommend when working to reduce stigma that EBPs are included, as illustrated in the proposed model by Papish et al. that targeted health care communities and included three main methods for reducing stigma attitudes: the right knowledge, professional balanced process, and a contact-based educational program [55]. It is important to highlight that even though these interventions could help in changing attitudes, they do not necessarily mean equal changes to the behavior of all individuals [5, 55, 81, 118, 128, 129]. Finally, we hope that this study will assist those in legislative and administrative positions and contribute to future campaigns against stigma of people with mental illness in Bahrain and across the region.

### Limitations

Our study did explore a large population of health care providers working in Bahrain; however, we still cannot generalize these findings because our sample selection was convenient and not randomized. Neither population was equally stratified and certain groups were overrepresented (mainly females and nurses); thus, our study has the potential of sampling bias. Other limitations are that Cronbach's alpha of the 15-item OMS-HC was found to be questionable, and we cannot exclude the possibility of the "burnout effect," which could have influenced the participants.

## Supporting information

**S1 Table. Brief literature review on the 15-item OMS-HC.** There are only 13 studies reported that used the 15-item OMS-HC. An overview of the participants, full scores, Cronbach's alphas, and some of the remarks and main conclusions are highlighted.
(PDF)

## Acknowledgments

The authors would like to thank the management at the psychiatric hospital and primary health centers for their help and collaboration in conducting this research at their facilities.

## Author Contributions

**Conceptualization:** Feras Al Saif, Hussain Al Shakhoori.

**Data curation:** Feras Al Saif, Hussain Al Shakhoori, Suad Nooh, Haitham Jahrami.

**Formal analysis:** Haitham Jahrami.

**Funding acquisition:** Feras Al Saif.

**Investigation:** Feras Al Saif, Suad Nooh.

**Methodology:** Haitham Jahrami.

**Project administration:** Feras Al Saif, Hussain Al Shakhoori, Suad Nooh.

**Resources:** Suad Nooh.

**Software:** Haitham Jahrami.

**Supervision:** Feras Al Saif, Hussain Al Shakhoori.

**Validation:** Feras Al Saif, Suad Nooh, Haitham Jahrami.

**Visualization:** Feras Al Saif.

**Writing – original draft:** Feras Al Saif.

**Writing – review & editing:** Feras Al Saif, Hussain Al Shakhoori, Suad Nooh, Haitham Jahrami.

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
