## [Decision Letter · Decision Letter 0]

28 Aug 2019

PONE-D-19-17947

Association between the attitude on stigma of mental illness and the attitude toward adoption of evidence-based practice within the healthcare providers in Bahrain

PLOS ONE

Dear Dr. Al Saif,

Thank you for submitting your manuscript to PLOS ONE. After careful consideration, we feel that it has merit but does not fully meet PLOS ONE’s publication criteria as it currently stands. Therefore, we invite you to submit a revised version of the manuscript that addresses the points raised during the review process.

We would appreciate receiving your revised manuscript by Oct 12 2019 11:59PM. To enhance the reproducibility of your results, we recommend that if applicable you deposit your laboratory protocols in protocols.io, where a protocol can be assigned its own identifier (DOI) such that it can be cited independently in the future. For instructions see: http://journals.plos.org/plosone/s/submission-guidelines#loc-laboratory-protocols

We look forward to receiving your revised manuscript.

Kind regards,

Lars-Peter Kamolz, M.D., Ph.D., M.Sc.

Academic Editor

PLOS ONE

Journal Requirements:

Reviewers' comments:

Reviewer's Responses to Questions

**Comments to the Author**

1. Is the manuscript technically sound, and do the data support the conclusions?

Reviewer #1: Yes

Reviewer #2: No

2. Has the statistical analysis been performed appropriately and rigorously? 

Reviewer #1: Yes

Reviewer #2: No

3. Have the authors made all data underlying the findings in their manuscript fully available?

Reviewer #1: Yes

Reviewer #2: Yes

4. Is the manuscript presented in an intelligible fashion and written in standard English?

Reviewer #1: No

Reviewer #2: No

5. Review Comments to the Author

Reviewer #1: Dear authors,

thank you for the opportunity to review the manuscript “Association between the attitude on stigma of mental illness and the attitude toward adoption of evidence-based practice within the healthcare providers in Bahrain”.

This is, in principle, a well-structured study on the basis of an excellent literature pool, whose basic statements could most probably apply to other countries.

The manuscript would profit from a reduction of the introduction / the materials and methods section (especially the measures section).

The citation style required by PLOS One is to be taken into consideration.

Reviewer #2: My main problem lies with the assessment and the method of assessment. I worry that the questionnaires were not sufficient to address the question the authors are asking. The use of English questionnaires is also questionable, especially given the many mistakes in the ms, not least the use of "stigmatized" to mean stigmatizing.

I don't think the statistical methods were appropriate to address the question and the large amount of data. I also do not think that the sample was representative of health professionals in Bahrain, oversampling nurses.

6. PLOS authors have the option to publish the peer review history of their article (what does this mean?). If published, this will include your full peer review and any attached files.

Reviewer #1: No

Reviewer #2: No

---

## [Author Response · Author response to Decision Letter 0]

30 Oct 2019

Reviewer 1: 

We reduced the introduction, material, and methods (especially the measures section). Standard English and the citations were corrected using professional service “Editage”. Thank you for your comments and suggestions. They were very helpful.

Reviewer 2: 

The introduction was modified according to the suggestions provided.

Language usage, spelling, and grammar editing service of the resubmitted manuscript was provided by a professional service “Editage”.

An Arabic version requires validation and it would not be applicable for those who are non-Arabic speakers healthcare providers in Bahrain. Our sample would be limited to locals and Arabs. English, on the other hand, is an official language in the country and the Bahraini healthcare system. We provided our contacts and ethical code in the questionnaire. We responded to all inquiries professionally and we assume that participants did understand the questions and respond correctly.

We added a response to this point in the manuscript (Measures). We chose to use the standardized 15-item version, as recommended in a study which examined the scale’s properties (Modgill, Patten et al. 2014).This scale was reduced to 15-items because of weak item-total correlation (below 0.20) found in four items and additional item was dropped after cross-loaded across all three factors equally; ending with the 15-item version.

A regression model was added in the manuscript with a new Table that summaries the findings. 

All healthcare providers (whether Bahrainis or non-Bahrainis) must pass English licensing exams and are communicating with co-workers or English speaking patients through English. In Bahrain, all Medical/Nursing universities, training programs for healthcare providers use English as the official language. The majorities are locals who are bilingual (Arabic, English) but (37.4%) as been reported in our sample are English speaking Expats (mainly from India, Philippines, and other countries). We agree with your point that was mentioned already in the ms(limitations section) that our sample has the potential for sampling bias, oversampling the females and nurses based on their high response and reachable participation. We added a further regression analysis for the data for your kind information, and out of the journal restrain further analysis of the data would be out of the scope of our aim of the study. Thank you for your comments and suggestions. They were very helpful.

---

## [Decision Letter · Decision Letter 1]

12 Nov 2019

Association between attitudes of stigma toward mental illness and attitudes toward adoption of evidence-based practice within health care providers in Bahrain

PONE-D-19-17947R1

Dear Dr. Al Saif,

We are pleased to inform you that your manuscript has been judged scientifically suitable for publication and will be formally accepted for publication once it complies with all outstanding technical requirements.

With kind regards,

Lars-Peter Kamolz, M.D., Ph.D., M.Sc.

Academic Editor

PLOS ONE

Additional Editor Comments (optional):

Reviewers' comments:

Reviewer's Responses to Questions

**Comments to the Author**

1. If the authors have adequately addressed your comments raised in a previous round of review and you feel that this manuscript is now acceptable for publication, you may indicate that here to bypass the “Comments to the Author” section, enter your conflict of interest statement in the “Confidential to Editor” section, and submit your "Accept" recommendation.

Reviewer #1: (No Response)

Reviewer #2: All comments have been addressed

2. Is the manuscript technically sound, and do the data support the conclusions?

Reviewer #1: (No Response)

Reviewer #2: Yes

3. Has the statistical analysis been performed appropriately and rigorously? 

Reviewer #1: (No Response)

Reviewer #2: Yes

4. Have the authors made all data underlying the findings in their manuscript fully available?

Reviewer #1: (No Response)

Reviewer #2: Yes

5. Is the manuscript presented in an intelligible fashion and written in standard English?

Reviewer #1: (No Response)

Reviewer #2: Yes

6. Review Comments to the Author

Reviewer #1: (No Response)

Reviewer #2: The authors addressed my concerns in their revisions. The ms reads well, the English has been improved dramatically. The introduction is now more focussed. The authors have also addressed my concerns regarding the analysis and the limitations with regard to potential sampling bias.

7. PLOS authors have the option to publish the peer review history of their article (what does this mean?). If published, this will include your full peer review and any attached files.

Reviewer #1: No

Reviewer #2: No

---

## [Editor Report · Acceptance letter]

21 Nov 2019

PONE-D-19-17947R1 

Association between attitudes of stigma toward mental illness and attitudes toward adoption of evidence-based practice within health care providers in Bahrain 

Dear Dr. Al Saif:

I am pleased to inform you that your manuscript has been deemed suitable for publication in PLOS ONE. Congratulations! Your manuscript is now with our production department. 

With kind regards,

on behalf of

Dr. Lars-Peter Kamolz 

Academic Editor

PLOS ONE